# Integrated microRNA–mRNA Expression Profiling Identifies Novel Targets and Networks Associated with Autism

**DOI:** 10.3390/jpm12060920

**Published:** 2022-06-01

**Authors:** Pritmohinder S. Gill, Harsh Dweep, Shannon Rose, Priyankara J. Wickramasinghe, Kanan K. Vyas, Sandra McCullough, Patricia A. Porter-Gill, Richard E. Frye

**Affiliations:** 1Department of Pediatrics, University of Arkansas for Medical Sciences, Little Rock, AR 72202, USA; srose@uams.edu; 2Arkansas Children′s Research Institute, Little Rock, AR 72202, USA; kkvyas@uams.edu (K.K.V.); mcculloughsandras@uams.edu (S.M.); portergillpa@archildrens.org (P.A.P.-G.); 3The Wistar Institute, 3601 Spruce St., Philadelphia, PA 19104, USA; dweeph@gmail.com (H.D.); priyaw@wistar.org (P.J.W.); 4Barrow Neurological Institute at Phoenix Children′s Hospital, Phoenix, AZ 85016, USA; rfrye@phoenixchildrens.com; 5Department of Child Health, University of Arizona College of Medicine, Phoenix, AZ 85004, USA

**Keywords:** autism spectrum disorder, lymphoblastoid cell lines, microRNA, mRNA, gene expression, RNA-seq, miRNA-seq

## Abstract

Autism spectrum disorder (ASD) is a complex neurodevelopmental disorder, with mutations in hundreds of genes contributing to its risk. Herein, we studied lymphoblastoid cell lines (LCLs) from children diagnosed with autistic disorder (*n* = 10) and controls (*n* = 7) using RNA and miRNA sequencing profiles. The sequencing analysis identified 1700 genes and 102 miRNAs differentially expressed between the ASD and control LCLs (*p* ≤ 0.05). The top upregulated genes were *GABRA4*, *AUTS2*, and *IL27*, and the top upregulated miRNAs were *hsa-miR-6813-3p*, *hsa-miR-221-5p*, and *hsa-miR-21-5p*. The RT-qPCR analysis confirmed the sequencing results for randomly selected candidates: *AUTS2*, *FMR1*, *PTEN, hsa-miR-15a-5p, hsa-miR-92a-3p*, and *hsa-miR-125b-5p.* The functional enrichment analysis showed pathways involved in ASD control proliferation of neuronal cells, cell death of immune cells, epilepsy or neurodevelopmental disorders, WNT and PTEN signaling, apoptosis, and cancer. The integration of mRNA and miRNA sequencing profiles by miRWalk2.0 identified correlated changes in miRNAs and their targets’ expression. The integration analysis found significantly dysregulated miRNA–gene pairs in ASD. Overall, these findings suggest that mRNA and miRNA expression profiles in ASD are greatly altered in LCLs and reveal numerous miRNA–gene interactions that regulate critical pathways involved in the proliferation of neuronal cells, cell death of immune cells, and neuronal development.

## 1. Introduction

Autism spectrum disorder (ASD) is an enigmatic neurodevelopment disorder that affects approximately 2% of children [1]. The etiology is not known in most cases, with studies suggesting that it involves multiple organs and physiological systems outside of the brain [2] and the deregulation of diverse subcellular molecular pathways [3]. For example, abnormalities in immune regulation; metabolism, including folate and energy metabolism; and redox regulations and methylation have been documented in multiple studies, but the origin of these abnormalities, how they are connected, and their effects on ASD symptoms remain unclear [4]. Molecular pathways including mTOR, PTEN, AKT, and β-catenin have also been found to be deregulated in ASD [3].

Lymphoblastoid cell lines (LCLs) provide a valuable tool to study underlying molecular regulation in ASD [5]. Biobanked LCLs have been utilized to study structural genetic changes, such as copy number variations and gene mutations; changes in cellular regulation, including mRNA [6] and microRNA (miRNA) expression [7]; metabolic alteration, including alterations in mitochondrial function [8], redox [9,10], and methylation metabolism [11]; and changes in immune regulation [12].

MicroRNAs (miRNAs) are small non-coding RNAs with tissue-specific expression that regulate post-transcriptional gene expression [13]. Given the diverse pathways that have been found to be dysregulated in ASD, studying miRNA expression may be very promising, as miRNAs can regulate many seemingly unrelated molecular pathways and may result in diverse changes in gene expression depending on the tissue. Some initial studies have found alternations in miRNAs in ASD using LCLs [7,14,15]. Despite promising results, studies remain with inconsistent findings. Since miRNAs regulate mRNAs, one approach is to integrate analysis of changes in both mRNA and miRNA expression to explore regulatory networks.

This study presents the mRNA and miRNA sequencing profile of control and ASD LCLs and an integrative analysis of miRNA–mRNA expression using miRWalk 2.0. The findings from the bioinformatic analysis show that miRNAs potentially regulate mRNAs involved in the proliferation of neuronal cells, the cell death of immune cells, epilepsy or neurodevelopmental disorders, and the WNT/β-catenin/PTEN signaling pathways.

## 2. Materials and Methods

Figure 1 shows the workflow of the investigation, showing the number of LCL samples used as controls and ASD groups and the steps for bioinformatic analysis of the sequencing data using miRWalk 2.0 and Ingenuity^®^ Pathway Analysis software (IPA^®^).

### 2.1. Materials

The following materials were procured from various vendors: RPMI 1640 culture media, penicillin/streptomycin, fetal bovine serum (FBS), phosphate-buffered saline (PBS), and the BCA Protein Assay Kit were all obtained from Thermo Fisher Scientific (Waltham, MA, USA). For qRT-PCR reagents and vendors were the miRNeasy Mini Kit (Qiagen, Valencia, CA, USA), RNeasy MinElute Cleanup Kit (Qiagen, Valencia, CA, USA), and miScript II RT kit (Qiagen, Valencia, CA, USA), SYBR green qRT-PCR assay (Qiagen, Valencia, CA, USA), and TaqMan assays (ThermoFisher Scientific, Carlsbad, CA, USA).

### 2.2. Cell Lines and Tissue Culture Conditions

Samples from children with autistic disorder were obtained from the Autism Genetics Resource Exchange (AGRE), a publicly available biomaterials repository located in Los Angeles, CA (AGRE; Los Angeles, CA, USA). AGRE collects samples from non-idiopathic cases of autism spectrum disorders. The procedures for screening children for collection are provided on the program website (https://www.autismspeaks.org/agre-program, accessed on 17 May 2022). Age and gender-matched controls with no documented behavioral or neurological disorder or a first-degree relative with a medical disorder were obtained from Coriell Cell Repository (Camden, NJ, USA).

LCLs were maintained in RPMI 1640 culture medium as previously described [7,8,10] in a humidified incubator at 37 °C with 5% CO_2_. These LCLs were used in a previous study examining bioenergetics [10] and miRNA in ASD [7]. In this manuscript, LCLs from subjects with autistic disorder (ICD-9: 299.0; ICD-10: F84.0) are designated as ASD and those from controls are designated as CNT (Table 1).

### 2.3. RNA and Small RNA Sequencing

#### 2.3.1. LCL RNA Sequencing (RNA-Seq)

Total RNA was extracted using Trizol reagent (ThermoFisher, Carlsbad, CA, USA) following the manufacturer′s procedure. The total RNA quantity and purity were analyzed by the Bioanalyzer 2100 and RNA 6000 Nano LabChip Kit (Agilent, Santa Clara, CA, USA) with an RIN number of >7.0. Approximately 10 ug of total RNA was subjected to isolated Poly (A) mRNA with poly-T oligo attached magnetic beads (ThermoFisher, Carlsbad, CA, USA). Following purification, the poly(A)- or poly(A)+ RNA fractions were fragmented into small pieces using divalent cations under an elevated temperature. The cleaved RNA fragments were reverse transcribed to create the final cDNA library in accordance with the protocol for the mRNA-seq sample preparation kit (Illumina, San Diego, CA, USA) and the average insert size for the paired-end libraries was 300 bp (±50 bp). Finally, paired-end sequencing was performed on an Illumina Hiseq 4000 following the vendor′s recommended protocol.

#### 2.3.2. LCLs Small RNA Sequencing (miRNA-Seq)

Total RNA was extracted using Trizol reagent (ThermoFisher, Carlsbad, CA, USA) following the manufacturer′s procedure. The total RNA quantity and purity were assessed with Bioanalyzer 2100 (Agilent, Santa Clara, CA, USA) with an RIN number of >7.0. Small RNA enrichment was performed by the excision of the 15 to 50 nt fraction from a polyacrylamide gel. Approximately 1 ug of enriched RNA was used to prepare the small RNA library, according to the protocol of the TruSeq Small RNA Sample Prep Kits (Illumina, San Diego, CA, USA). Finally, single-end sequencing (36 bp or 50 bp) was performed on an Illumina Hiseq 2500 at LC Sciences (Houston, TX, USA) following the vendor′s recommended protocol.

### 2.4. Bioinformatics Analysis for RNA-Seq

The raw reads of both sequencing profiles were aligned using Bowtie [16] against the hg19 version of the human genome, and RSEM v1.2.12 software [17] was used to estimate raw read counts using Ensemble v84 gene information. DESeq2 [18] was utilized to identify differentially expressed genes between sample groups. Samples with poor alignment rates were identified as outliers in quality control analysis and therefore removed before carrying out the differential expression analysis. A false-positive rate of α = 0.05 with false-discovery rate (FDR) correction was used as the level of significance. Only a handful of genes were found to satisfy the FDR < 5% cut-off, which was not sufficient for functional enrichment analysis. Therefore, it was decided to consider a *p* < 0.05 threshold to select differentially expressed genes. These genes were then subjected to functional enrichment analysis.

### 2.5. Functional Annotation of Differentially Expressed Genes

Pathway and functional enrichment analyses were tested on genes that passed a significant *p*-value of <0.05 using QIAGEN’s Ingenuity^®^ Pathway Analysis (IPA^®^, QIAGEN Redwood City, CA, USA) software. The most relevant (*p* < 0.05) pathways and functions associated with autism were selected to generate dot plots using a customized R script.

### 2.6. miRNA Target Predictions using miRWalk2.0

The miRWalk2.0 database was used to gather putative miRNA binding sites within the DEGs [19,20]. miRWalk3.0 is a new publicly available version of miRWalk2.0. However, miRWalk3.0 is not very useful for carrying out miRNA-target predictions, as it fails to offer users the flexibility to choose the prediction algorithm of their choice, and at the same time, it entirely misses all the key features that are of utmost importance to the scientific community (e.g., a meta-analysis of targets by 13 different prediction datasets). On the other hand, the miRWalk2.0 database offers a meta-analysis of putative miRNA binding sites by collecting 13 prediction datasets from existing miRNA-target resources, which can help reduce the number of false-positive targets [21,22].

### 2.7. Integrated Analysis of RNA- and miRNA-Sequencing Data

After the initial analysis of removing bad quality reads, the DESeq2 package [18] was utilized to normalize the raw counts and fit models to identify differentially expressed genes (DEGs) and miRNAs (DEMs)between ASD and control samples. The level of significance was set to *p* < 0.05. The meta-analysis platform of the miRWalk2.0 database [19,21] was employed for this integrated analysis to collect putative interactions between significant genes and miRNAs. The interactions among significant genes and miRNAs predicted with at least 2 algorithms were compiled into a list (only DEGs) and were uploaded to IPA for enrichment analysis and identification of associated networks (Figure 1).

### 2.8. Quantitative Reverse-Transcriptase-Polymerase Chain Reaction (qRT-PCR) Validation

LCLs were used to isolate total RNA and microRNA as previously described [7]. Briefly, RNA isolated with miRNeasy Mini Kit (Qiagen, Valencia, CA, USA) and miRNA isolated with RNeasy MinElute Cleanup Kit (Qiagen, Valencia, CA, USA) were used to perform cDNA synthesis with the miScript II RT kit (Qiagen, Valencia, CA, USA). The qRT-PCR was run in triplicate on a QuantStudio™ 6 Flex Real-Time PCR System (ThermoFisher Scientific, Carlsbad, CA, USA).

Appendix A shows the assay ID and catalog number for the TaqMan gene expression assay (ThermoFisher Scientific, Inc., Carlsbad, CA, USA) for *AUTS2, FMR1,* and *PTEN*. SYBR green miScript assays (Qiagen, Valencia, CA, USA) for miRNA expression were used for *Hsa-miR-15a-5p, Hsa-miR-92a-3p*, and *Hsa-miR-125b-5p* (Appendix A). Taqman assays were duplexed with *GAPDH* to normalize the mRNA expression. miRNA data normalization was conducted with an exogenous control (Ce-miR-39) and an endogenous control (*RNU6*).

Negative controls and no template controls (NTC) were run with each assay. Relative quantitation for miRNA and mRNA was calculated using the 2^-ΔΔCt^ method.

### 2.9. Statistical Analysis

All experimental data for qRT-PCR are presented as means ± SEM, and differences between the two groups were examined using Student′s *t*-test (2-tailed). *p*-Values of less than 0.05 were considered significant.

## 3. Results

### 3.1. Identification of Differentially Expressed Genes and miRNAs

The top candidate genes and miRNAs are shown in Figure 2 and Figure 3. The samples with poor alignment rates were dropped from the subsequent analysis. These results demonstrated that genes and miRNAs were differentially expressed between ASD and control samples. A false-positive rate of α = 0.05 with false discovery rate (FDR) correction was used as the level of significance.

#### 3.1.1. Differential Gene Expression in ASD LCLs

The RNA-seq analysis showed 1700 significantly (*p* < 0.05) differentially expressed mRNAs (DEGs) between the ASD and control groups (Figure 2A). The list of deregulated genes shows that 629 were downregulated and 1071 were upregulated (Appendix A).

The top 45 (25 most upregulated and 20 most downregulated) differentially regulated genes (FDR < 5%) in ASD LCLs compared with control LCLs are depicted in Figure 2B. ASD LCLs showed highly upregulated expression for *GABRA4*, with a fold change of 607 (*p* ≤ 0.05), followed by *IL27*, with a fold change of 27.7 (*p* ≤ 0.05), compared with control LCLs. By contrast, the downregulated genes (*p* < 0.05) were from immunoglobulin light chain and heavy chain members in ASD LCLs, e.g., *IGKV1-6*, with a fold change of -4127.6, followed by *IGLV1-40*, with a fold change of -3750 (*p* ≤ 0.05).

#### 3.1.2. Differentially Expressed miRNAs in ASD LCLs

The small RNA-seq analysis (miRNA-seq) showed 102 differentially expressed miRNAs (DEMs) in ASD compared with control samples (*p* < 0.05) (Figure 3A; Appendix A). Of these, 37 miRNAs were downregulated, and 65 were upregulated.

The top 29 (18 most upregulated and 11 most downregulated) differentially regulated miRNAs (FDR < 5%) in ASD LCLs compared with control LCLs are depicted in Figure 3B. ASD LCLs showed significant upregulation for a number of miRNAs (*p* < 0.05) compared with control LCLs. These included hsa-miR-6813-3p, hsa-miR-221-5p, and hsa-miR-21-5p, with fold changes of 13, 6.2, and 3.6, respectively. The top downregulated miRNAs (*p* < 0.05) were *hsa-miR-150-5p*, hsa-*miR-874-5p*, and *miR-15b-5p*, with fold changes of −5.7, −4, and −1.6, respectively.

### 3.2. qRT-PCR Validation of Differentially Expressed miRNAs and Genes

To verify the findings of both sequencing results, we randomly selected three differentially expressed genes and three differentially expressed miRNAs for qRT-PCR validation.

#### 3.2.1. Gene Expression Validation by qRT-PCR

To validate the RNA-seq dataset, the expression levels of the three randomly selected genes *AUTS2, FMR1,* and *PTEN* were determined by qRT-PCR analysis. The expression levels of AUTS2 (Figure 4A) were upregulated 3.7-fold in ASD LCLs compared with the control LCLs (*p* ≤ 0.05). The other two mRNAs, *FMR1* (Figure 4B) and *PTEN* (Figure 4C), were significantly downregulated in ASD LCLs compared with the control LCLs (*p* ≤ 0.05).

#### 3.2.2. miRNA Expression Validation by qRT-PCR

To validate the miRNA-seq dataset, the three miRNAs, *hsa‐miR‐15a‐5p, hsa‐miR‐92a‐3p, and hsa‐miR‐125b‐5p p* (Figure 3 and Appendix A), were randomly chosen for qRT-PCR analysis. The expression levels of *miR-15a-5p* (Figure 5A) were downregulated 2.2-fold in ASD LCLs compared with the control LCLs (*p* ≤ 0.05). The other miRNA *miR-92a-3p* (Figure 5B) was also significantly downregulated in ASD LCLs compared with the control LCLs (*p* ≤ 0.05). By contrast, *miR-125b-5p* (Figure 5C) was significantly upregulated in ASD LCLs compared with the control LCLs (*p* ≤ 0.05). These results confirm the miRNA-seq dataset for the differentially expressed miRNAs in ASD LCLs.

### 3.3. Pathway Analysis of Differentially Expressed Genes

A total of 1700 DEGs and 102 DEMs with *p* < 0.05 were selected to test for pathway enrichment analysis using IPA software. The significantly expressed genes can participate in a network of diverse signaling pathways, such as apoptosis, cell death of immune cells, cell cycle progression, epilepsy, nNOS, and proliferation of neuronal cells (Figure 6). Z-scores are represented by red or blue colors for the signaling pathways. The activated pathways (shown in red) were related to cell death of immune cells, inflammatory response, the proliferation of neuronal cells, central nervous system solid tumor, epilepsy, the endocannabinoid neuronal synapse pathway, and synaptic long-term potentiation. Inhibited pathways (shown in blue) controlled apoptosis, inflammation of the organs, movement disorders, motor dysfunction, PI3K/AKT, ATM, and PTEN (Appendix A).

### 3.4. Functional Enrichment Analysis of Differentially Expressed Genes

Subsequently, the 1700 DEGs and 102 DEMs (*p* < 0.05) were further selected to test for functional enrichment analysis using IPA software. Interestingly, these genes are also associated with diverse functions related to cancer, apoptosis, cellular homeostasis, movement disorders, and proliferation of neuronal cells (Figure 7). The genes with activated functions (shown in red) regulate cancer, cell survival, cell death of immune cells, neuronal cell death, the endocannabinoid neuronal synapse pathway, and synaptic long-term potentiation. By contrast, genes with inhibited functions (shown in blue) were involved in apoptosis, movement disorders, motor dysfunction, seizure, colon cancer, ATM signaling, PI3K/AKT signaling, and neuronal cell death. For details on various functions associated with ASD and control LCLs, see Appendix A.

### 3.5. Integrated Analysis of miRNA and mRNA Expression

The mRNA–miRNA interaction analysis using miRWalk2.0 showed 267 genes in significant pathways that were predicted to be targeted by deregulated miRNAs (Appendix A). These pathways control cell death of immune cells, the proliferation of neuronal cells, epilepsy or neurodevelopmental disorders, and Wnt/β-catenin and PTEN signaling (Appendix A). The activated pathways are shown in red, whereas the inhibited pathways are shown in blue.

The ASD LCLs showed upregulation of *GABRA4*, *IL27*, and *PTEN*; the downregulated genes were *FOXP1*, *NTN1*, and *NCAM2* (Figure 8 and Appendix A). For example, *GABRA4* was upregulated 606-fold in ASD LCLs and regulated pathways of epilepsy and neurodevelopmental disorders (Figure 8, Appendix A). *GABRA4* was predicted to be targeted by 16 upregulated miRNAs and 10 downregulated miRNAs (Figure 8, Appendix A). By contrast, only *miR-3529-3p* (Appendix A) was validated to be the target of *GABRA4*. Appendix A also shows that *miR-3529-3p* was upregulated twofold (red) in ASD LCLs, and this has been validated by two publications. *PTEN* regulates pathways of cell death of immune cells and proliferation of neuronal cells. *PTEN* was targeted by *miR-21-5p*, which was upregulated fourfold in ASD LCLs, and this has been validated by 62 publications (Appendix A). *miR-3529-3p* also targets *NCAM2*, which was upregulated 6.06-fold (Figure 8). The sequencing results showed that *miR-3529-3p* was upregulated twofold in ASD LCLs, and only one publication validates this result (Appendix A).

ASD LCLs show the pathways of cell death of immune cells are also activated by *IL27*, *PTEN*, and *FOXP1* (Figure 8), whereas *PTEN, NTN1*, and *NCAM2* regulate the pathway of the proliferation of neuronal cells.

## 4. Discussion

The RNA-seq analysis in our study showed 1700 DEGs in ASD LCLs compared with control LCLs. Most notably, *GABRA4*, *KREMEN1*, *PLXNA1* and *IL27* were elevated in ASD LCLs compared with control LCLs, whereas the majority of significantly downregulated genes were from immunoglobulin family members (*IGLV1-40*, *IGKV1-6*, and *IGHV3-13*) (Figure 2). Of the 102 DEMs, this study revealed that numerous miRNAs, notably *miR-6813-3p*, *miR-221-5p*, *miR-3529-3p*, *miR-1271-5p* and *miR-21-5p* were upregulated, whereas *miR-150-5p*, *miR-874-5p*, *miR-940*, *miR-17-3* and *miR-33b-3p* were markedly downregulated (Figure 3) in the ASD LCLs compared with control LCLs.

### 4.1. Dysregulated Expression at Transcriptomic and Epigenetic Levels in ASD

Previous transcriptomic [6,14,15,23,24,25,26,27,28] and epigenetic [14,15,23] studies on LCLs in ASD help us understand the genes and their functional pathways involved in the risk of developing ASD. For example, Talebizadeh et al. 2014 [24] looked at alternate splicing in ASD using LCLs and identified *CYFIP1*, *ZMYM6*, and *TRAP150* as potential ASD candidate genes. The RNA-seq dataset from the current study could not detect *TRAP150* and showed no significant change for *CYFIP1* and *ZMYM6*. Hu and colleagues [25] showed that the *NFKB1* and *MBD2* to be a strong candidate for genes for ASD, and the RNA-seq data on ASD LCLs did not detect expression of *MBD2* or *NFKB1*. Mutations in the fragile X mental retardation 1 gene (*FMR1*) are associated with the inherited form of ASD, and Nishimura and colleagues [26] found downregulation of *FMR1* in ASD vs. control samples, and our study confirmed this with qRT-PCR (*p* < 0.05) (Appendix A and Figure 5). However, for the *JAKMIP1* and *GPR155* genes, which were differentially expressed in their study [26], our RNA-Seq ASD LCLs data show no significant change for *JAKMIP1* and *GPR15* (Appendix A).

Significant expression of *HEY1*, *SOX9*, *miR-486-3p/5p*, and *miR-181c* was observed in ASD LCLs [23], but our current RNA-seq dataset results did not reach significant level for these genes. *MiR-146* was upregulated in ASD LCLs [15], and the current miRNA-seq dataset confirmed the significant upregulation of *miR-146b-5p* (FC = 1.99; *p* = 0.006) (Appendix A). Another study on ASD LCLs [14] reported upregulation of *miR-29b* and downregulation of *miR-199b*. The miRNA-seq data showed upregulation of *miR-199b-5p* and *miR-199b*-3p with no significant *p*-value, whereas *miR-199a-5p* showed significant upregulation in ASD LCLs (FC = 5.8, *p* = 0.0065). For *miR-29b*, our data showed downregulation, but the results did not reach statistical significance.

ASD LCLs showed significant upregulation of *GABRA4* (Figure 2). An association involving *GABRA4* has been reported in ASD patients [27,28], and this is the major inhibitory neurotransmitter in the mammalian brain. *GABRA4* is expressed in the thalamus, striatum, cerebral cortex, dentate gyrus (DG), and CA1 region of the hippocampus [29]. Another well-known gene in autism is *AUTS2*, and this study reported upregulation of *AUTS2* expression in ASD LCLs compared with control LCLs (Figure 2, Figure 4A). Most studies have reported intragenic de novo deletions of *AUTS2* [30,31,32] and have rarely described disease point mutations [33].

Many studies using LCLs as model cell type to understand autism, including this one, have many limitations, particularly the modest sample sizes in the context of the heterogeneous nature of ASD. Indeed, ASD is defined by a collection of various symptoms, which can be different from patient to patient. Although some categories are used as modifiers of the diagnosis, such as with and without language impairment or intellectual disability, the variation in symptoms has been difficult to easily describe or categorize. Thus, without well-defined subgroups, it may be difficult to find consistency with modest sample sizes. Clearly, larger studies on well-defined groups of individuals with ASD will be needed in the future. However, despite the heterogeneous nature of ASD, this study was consistent with some previous studies.

### 4.2. Immune Abnormalities and ASD

Immunoglobulin polymorphisms are known for infection and auto-immune disease susceptibility. Dysfunction of the immune system appears to be associated with ASD. A recent meta-analysis demonstrated evidence for immunological dysregulation in ASD with a reduction in total IgG and an elevation in the IgG4 subset [34]. In addition, studies have demonstrated that total IgM and IgG concentrations correlate with aberrant behavior with lower immunoglobulin concentrations being associated with worse behavior [35]. Additionally, a meta-analysis also demonstrated that intravenous immunoglobulin improved aberrant behavior, specifically irritability, hyperactivity, and social withdrawal [34]. Thus, this clinical data is consistent with the findings of decreased expression of genes involved in the production of IgG.

IL-27 is a pleiotropic cytokine involved in infection, cellular stress, neurological disease, and cancer that has complex activating and inhibitory properties in both innate and acquired immunity [36]. It is secreted from and binds to microglia, macrophages, and neurons and promotes neuronal survival by regulating cytokines, neuroinflammation, oxidative stress, apoptosis, autophagy, and epigenetics [36].

*IL-27* gene expression was found to be markedly elevated in ASD LCLs. In the BTBR mouse model of ASD, studies have demonstrated that IL-27 agonist normalized neuroimmune dysfunction [37] yet other studies show that methylmercury [38] and tyrosine kinase inhibitor tyrphostin AG126 [39] decrease IL-27 in the BTBR mouse model. Another study demonstrated decreased IL-27 production by CD14+ cells derived from ASD individuals following in-vitro immunological challenge [40].

### 4.3. Integrated Analysis of ASD-CNT LCLs

Following, RNA-Seq and functional enrichment, an integrated analysis recognized a series of potential miRNA-mRNA interactions with implications in regulation of diverse networks. . For example, *PTEN* was shown to be downregulated by RNA-seq analysis (Appendix A) and RT-qPCR (Figure 4C). *PTEN* silencing enhances neuronal proliferation and differentiation by activating the PI3K/Akt/GSK3β pathway [41]. PTEN regulates the pathways for cell death of immune cells and the proliferation of neuronal cells (shown in red). Appendix A also shows that *PTEN* was predicted to be targeted by 15 upregulated miRNAs, such as *miR-221-5p*, *miR-21-5p*, *miR-148b-3p*, and *miR-26b-5p*, and there were five validated targets: *miR-21-5p*, *miR-152-3p*, *miR-26b-5p*, *miR-103a-3p*, and *miR-107*. For example, miR-21-5p was upregulated in ASD LCLs fourfold (Appendix A). *GABRA4* showed a 606.95-fold upregulation in ASD LCLs (Figure 2, Appendix A) and is a target for *miR-3529-3p*. A role of *GABRA4* has been shown in ASD and seizure susceptibility [42,43]. The integrated analysis showed that it regulates the pathway for epilepsy or neurodevelopmental disorders; it is predicted to be targeted by 10 miRNAs: *miR-150-5p*, *miR-874-5p*, *miR-33b-3p*, *miR-940*, *miR-17-3p*, *miR-324-5p*, *miR-766-3p*, *miR-484*, *miR-197-3p*, and *miR-342-3p*. Similarly, the upregulated gene IL27 (Figure 2, Appendix A) plays an important neuroprotective role [25], and the integrated analysis showed that it is important in cell death of immune cells, has one predicted target (miR-7-5p), and has no validated miRNA. Multiple studies have converged on the dysregulation of the Wnt/β-Catenin pathway in association with ASD [44], including studies using mouse [45], Drosophila, and zebrafish [46] animal models and patient cases [47].

The integrated analysis of miRNA–mRNA showed that 29 miRNAs (18 upregulated and 11 downregulated) and 267 genes formed miRNA–target gene pairs, which may have a role in complex network to regulate the proliferation of neuronal cells, cell death of immune cells, epilepsy or neurodevelopmental disorders, and Wnt/β-catenin and PTEN signaling (Figure 8). Appendix A represents new mediators of abnormal gene expression that could be potential targets for further exploration and therapeutic interventions in the proliferation of neuronal cells, epilepsy or neurodevelopmental disorders, and Wnt/β-catenin and PTEN signaling. However, functional validation is needed to test our miRNA–mRNA integration findings in ASD LCLs.

### 4.4. Autism and Cancer Genes Link?

Some have observed an overlap between the molecular pathways in ASD and cancer [48]. In this study, we found many cancer genes that demonstrated changes in both mRNA and miRNA expression in the ASD LCLs. The ASD LCLs demonstrated the downregulation of mRNA of several tumor suppressor genes, including *KREMEN1* [49], *ST5* [50], *ILDR1* [51], and *FOXP4* [52].

The ASD LCLs demonstrated upregulation of several oncogenes, including *SYK* [53,54,55], *MFHAS1* [56,57,58,59], and *TCL1A* [60]. Interestingly, *MFHAS1* has also been implicated in sepsis-associated encephalopathy [61] and intellectual impairment [62]. By contrast, *SSBP2*, a gene involved in DNA and telomere repair and growth arrest of cancer cells [63] was also found to be upregulated in ASD LCLs.

Several miRNAs involved in cancer show dysregulated expression in ASD LCLs, including *miR-6732-3p*, *hsa-miR-221-5p* [64,65], *hsa-miR-21-5p* [66,67], *hsa-miR-152-3p* [68,69], *hsa-miR-3529-3p* [70,71], and *has-miR-148b-3p* [72,73]. Additionally, *miR-3529-3p* was 2-fold upregulated in ASD LCLs and it targets *GABRA4* (Appendix A) and this miRNA promotes angiogenesis [71]. Studies have shown *miR-3529-3p* to be upregulated in radiotherapy-resistant colorectal cancer cells [70] and downregulated in liver cancer [74].

## 5. Conclusions

RNA-Seq analysis identified top 45 genes and 29 miRNAs that were differentially expressed in ASD LCLs compared to control LCLs. An integrated miRNA–mRNA analysis showed that the genes in significant pathways that are predicted to be targeted by deregulated miRNAs control apoptosis, cell death of immune cells, cell cycle progression, epilepsy, and proliferation of neuronal cells. Our results reinforce findings from other groups in regard to important underlying molecular pathways and processes, such as the regulation of cell growth, the role of organs besides the brain in ASD (e.g., the immune system), and the overlap with other non-neurodevelopmental diseases, such as cancer. These findings refine the landscape of ASD genes using LCLs and show the importance of studying different cell types to understand the regulatory networks in ASD.

## Figures and Tables

**Figure 1 jpm-12-00920-f001:**
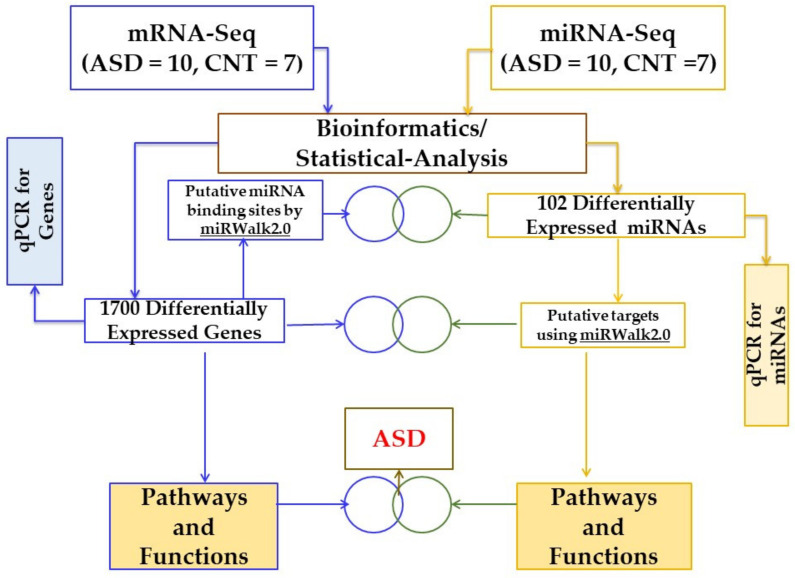
Schematic workflow to study differentially regulated genes, miRNAs, and pathways in ASD and CNT LCLs. ASD = autism, CNT = control, LCLs = lymphoblastoid cell lines.

**Figure 2 jpm-12-00920-f002:**
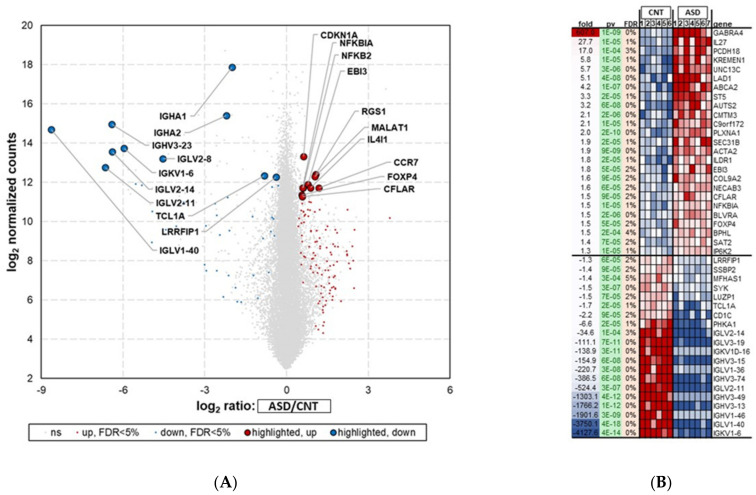
RNA-seq analysis showing differential expression of top 45 mRNAs in LCL groups—ASD (*n* = 7) and CNT (*n* = 6)—with FDR < 5%. Samples with poor alignment rates which were dropped from differential expression analysis, included 3 from ASD group and one from Control group. (**A**) Volcano plot of differentially expressed mRNAs (DEGs) in ASD and CNT LCLs. (**B**) Heatmap of DEGs in ASD and CNT LCLs. ASD = Autism, CNT = Control, Fold = Fold Change, pv = *p*-value and gene symbol. Red dots are upregulated, blue dots are downregulated, and grey dots indicate no change; FDR = false discovery rate.

**Figure 3 jpm-12-00920-f003:**
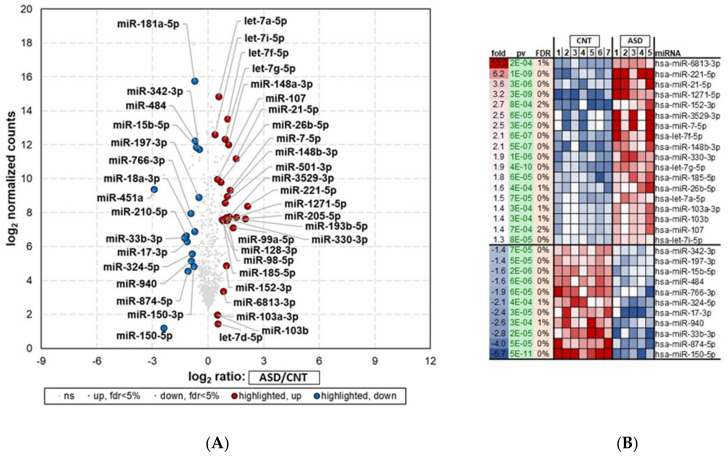
miRNA-seq analysis showing differential expression of top 29 miRNAs in LCLs groups—ASD (*n* = 5) and CNT (*n* = 7)—with FDR < 5%. Samples with poor alignment rates which were dropped from differential expression analysis, included 5 from ASD group. (**A**) Volcano plot of differentially expressed miRNAs in ASD and CNT LCLs. (**B**) Heatmap of differentially expressed miRNAs in ASD and CNT LCLs. ASD = Autism, CNT = Control, Fold = Fold change,, pv = *p*-value, and miRNA. Red dots are upregulated, blue dots are downregulated, and grey dots indicate no change; FDR = false discovery rate.

**Figure 4 jpm-12-00920-f004:**
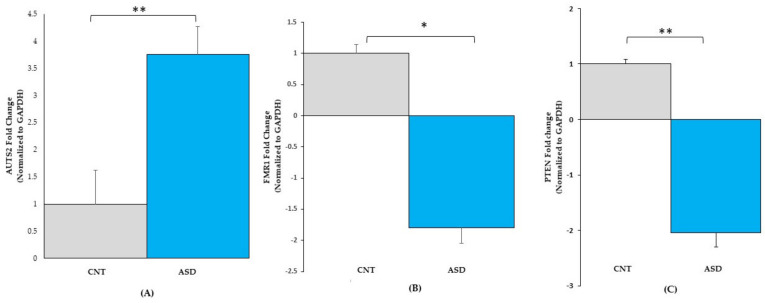
qRT-PCR validation of gene expression in LCL groups for (**A**) AUTS2: CNT (*n* = 4), ASD (*n* = 4), (**B**) FMR1: CNT (*n* = 7), ASD (*n* = 10) and (**C**) PTEN: CNT (*n* = 7), ASD (*n* = 10). ASD = Autism, CNT = Control, * = *p* < 0.05, ** = *p* < 0.0001; error bars represent standard error of the mean.

**Figure 5 jpm-12-00920-f005:**
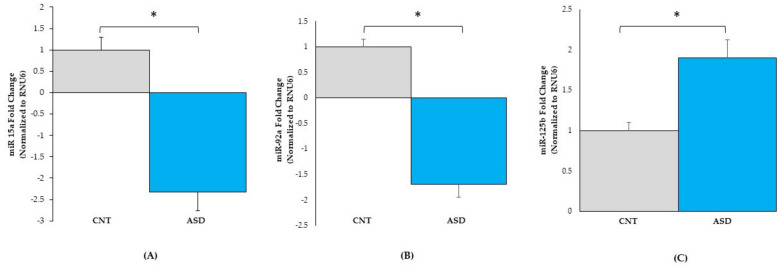
qRT-PCR validation of miRNA expression in control (*n* = 7) and ASD (*n* = *10*) LCLs. *miR-15a-5p* (**A**), *miR-92a-3p* (**B**), and *miR-125b-5p* (**C**). ASD = autism, CNT = control; * = *p* < 0.05 compared to controls; error bars represent standard error of the mean.

**Figure 6 jpm-12-00920-f006:**
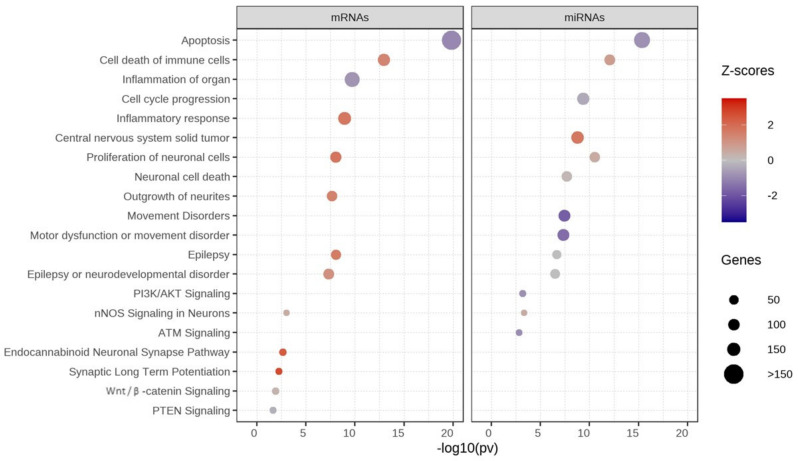
Pathway enrichment analysis of significant mRNAs and miRNAs-Ingenuity Pathway Analysis (IPA). Z-scores show activation or inhibition of pathways associated with autism. Red = activated pathway, blue = inhibited pathway, size of circle = number of genes.

**Figure 7 jpm-12-00920-f007:**
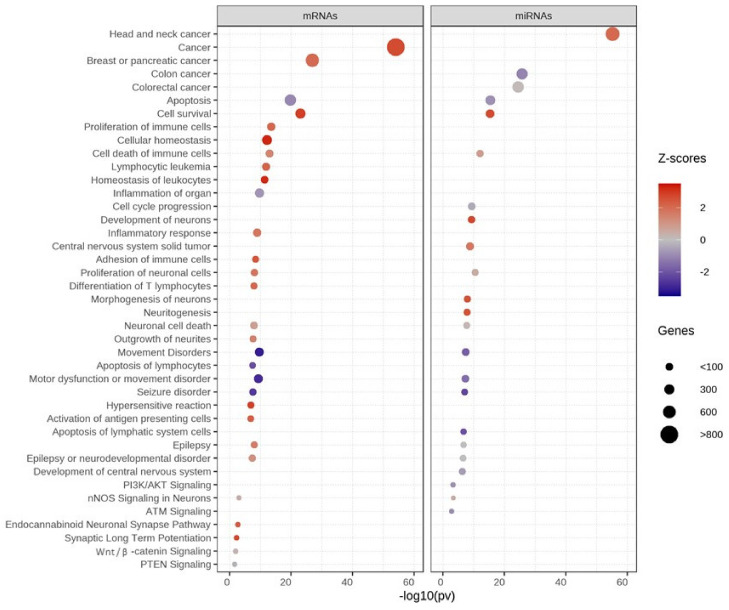
Functional enrichment analysis of significant mRNAs and miRNAs using IPA. Z-scores show functional enrichment. Red = activated, blue = inhibited, size of circle = number of genes.

**Figure 8 jpm-12-00920-f008:**
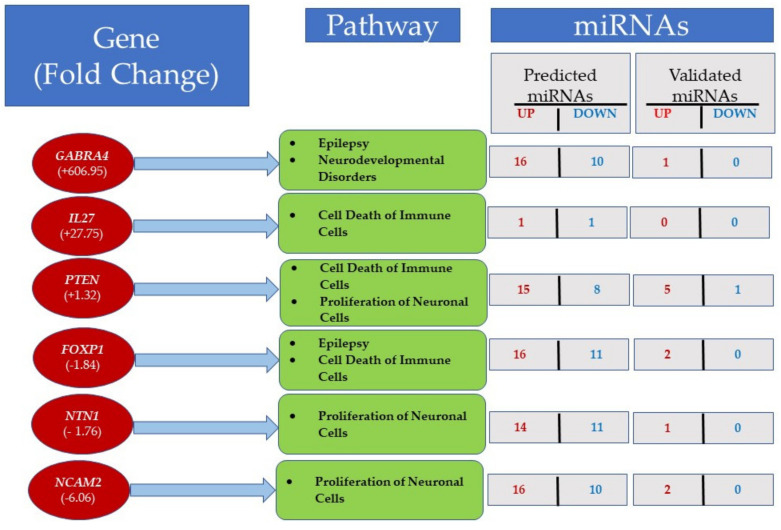
mRNA–miRNA interactions were predicted using differentially expressed genes targeted by miRNAs with miRWalk2.0. The figure shows some important pathways in LCLs, along with the number of predicted and validated miRNAs. For details see Appendix A.

**Table 1 jpm-12-00920-t001:** Lymphoblastoid Cell Lines (LCLs) used in this study from children with autistic disorder and age- and gender-matched control LCLs. Average age of each group is provided.

Unrelated Controls (CNT)		Study Sample Number (CNT)	Autism (ASD)		Study Sample Number (ASD)
ID	Age		ID	Age	
GM09621	8	09621C	03C14441	7	14441A-A
GM16007	12	16007C	03C16499	11	16499A-A
GM15862	11	15862C	AU0939303	11	2591A-A
GM11626	13	11626C	AU1165302	13	2746A-A
			AU1280302	7	3110A-A
			AU1393306	3	3540A-A
GM11599	9	11599C	AU1344302	7	3620A-A
GM09659	4	09659C	01C08495	4	8495A-A
			01C08594	7	8594A-A
GM10153	10	10153C	02C09713	10	9713A-A
Average (SD)	9.6 (3.0)			8.0 (3.2)	

## Data Availability

Data are available upon request.

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
