# Peer review of "Integrated microRNA–mRNA Expression Profiling Identifies Novel Targets and Networks Associated with Autism"

_jpm, 2022, doi:10.3390/jpm12060920_

Round 1

Reviewer 1 Report

  • Figure 1 should be moved from "introduction" section to "materials and methods".
  • How did you calculate sample size?
  • Authors should describe about inclusion criteria of cases and selection of controls.
  • Would assume that there are two "2.5" and "2.6" sections?

Author Response

  1. Figure 1 should be moved from "introduction" section to "materials and methods".

Thank you for this suggestion, we have moved figure 1 to materials and methods section. See lines 73-80.

Also to improve methods description, we have imoved the supplementary methods section into the main body of manuscript. See lines 108-149.

  1. How did you calculate sample size?

We did not use power analysis to estimate the sample size. We’re aware with the fact that by increasing sample size or sequencing depth increases power. Moreover, hundreds of RNA-seq studies have been published with just 3 vs 3 experimental design and in our manuscript, we have used miRNA-seq [] and mRNA-seq []. Also, herein the rational was more on the budget for propagation and analysis of LCLs. Due to the cost of RNA-seq experiments and our limited budget, we were only able to use limited samples for sequencing work. Also other studies using LCLs also have had small sample sizes, e.g., Sarachana, T., et al., Genome Med, 2010. 2(4): p. 23.; Talebizadeh, Z., et al  Autism Res, 2008. 1(4): p. 240-50; Talebizadeh, Z., R. et al  Psychiatr Genet, 2014. 24(1): p. 1-9.

  1. Authors should describe about inclusion criteria of cases and selection of controls.

The samples were obtained from the Autism Genetics Resource Exchange (AGRE), a publicly available biomaterials repository. This repository collects samples from children with idiopathic autism. The website referring to their screening procedures and inclusion criteria is now provided in the manuscript.  See Lines 88-91.

  1. Would assume that there are two "2.5" and "2.6" sections?

Thank you for letting us know this mistake, we have corrected this error and revised the numbering in this section. See lines 151-198.

Reviewer 2 Report

Gill et al have used mRNA and miRNA sequencing to determine DEG in autism lymphoblastoid cells from patients. After random validation (3 mRNAs, 3 miRNAs), pathway and functional analysis, the authors look at integrated analysis using miRWalk2.0. Overall, the data derived from this work is of interest to the field and would be suitable for publication in JPM. However, there a few major concerns and many minor concerns this reviewer has before this manuscript could be in an acceptable format.

Major concerns:

  • This ties to many minor concerns. Although overall, the MS is an okay read, it is plagued by typos, syntax errors, and spelling mistakes. There are too many to mention in the minor concerns so the authors need to carefully edit the MS before resubmission.
  • Scientifically, the big issue I have is the crux of the novelty of this paper is tying or correlating changes in miRNA to the changes in mRNA observed in autism LCLs. Yet this is a footnote in the results tied to a supplemental table. The authors need to provide a main figure that describes the integrated analysis findings, if even for the most consistent correlations between miRNA changes and mRNA changes.
  • Related to point 2. I am not buying randomly validating 3 hits. These could be supplemental if adding the above figure is an issue. The hits that should be validated should be based on the integrated analysis to validate the 3 most convincing miRNA – mRNA gene pairs.
  • There is a wide inconsistency throughout the paper when referring to AKT/PTEN/WNT pathways:

“PI3K/AKT, ATM and PTEN”

“ATM signaling, PI3K/AKT signaling”

“PI3K/Akt/GSK3β pathway”

“WNT/PTEN signaling”

Please be consistent throughout and see spelling issues in minor concerns.

  • The authors spend a large focus on section 4.4 and cancer genes. The section is fine but please heavily condense and make more concise.
  • Wnt is an important pathway in neurodevelopmental disorders/autism. The authors should cite some literature showing increased and decreased Wnt is ASD related genes/conditions (see. PMID: 31981384, PMID: 35044823, PMID: 28402856).
  • The beginning of the discussion, rightfully, compares the authors results to other expression analysis of ASD cells. The authors need to comment in section 4.1 on the overall discrepancies and potential limitations of performing these types of experiments.
  • In section 4.3 lines 328 to 331, the authors are referring to results, even stating shown in read without referring to a figure. Please re-write this section more clearly.

Minor concerns:

- Every gene mentioned needs to be italicized, unless you are talking about protein level, which I don’t think the authors ever are.

-Page 2 lined 45 – “beta-catherin”, I assume should read beta-catenin, I also recommend using β if allowed by the journal as well as being consistent in later instances

- this reviewer does not understand the repeated instances of “Epilepsy / epilepsy”. Please either correct, explain, or simplify with “seizure-disorders”. This shows up a few times.

-Page 3 line 76, remove space between CA and closed bracket.

-Page 3 line 77, first word should be plural, “Samples”

- Page 3 line 82, spacing issues after “were”

-Page 4 line 131, “. (Fig. 1).” Remove first period, and other figures are referred as “Figure X”. Please make consistent.

- Page 6 line 180, remove space between Figure 3 and open bracket.

- Page 7 line 202, please unitalicized “and”.

- figure 6, I am unsure what Wnt/I2-catenin signaling is. Should read Wnt/β-catenin. I also recommend this to be used for most instances in the paper.

- Page 10 line 256, fix double periods

- Page 10 line 258, fix “signaliing” to signaling

- Page 11 line 286, fix spacing issue after “results”

- Page 11 line 298, fix “upregulaation” to upregulation

- Page 11 line 299, fix “cntrol” to control

- Page 11 line 307 and 310, fix “abarrent” to aberrant

- Page 12 line 313, assuming “th” should read “that”?

- Page 12 line 318, please explain or expand BTBR

- Page 12 line 318, assuming “mode” should read “model”?

Again, this is not all the issues, there are more, please spell and grammar check closely before resubmitting.

Author Response

Gill et al have used mRNA and miRNA sequencing to determine DEG in autism lymphoblastoid cells from patients. After random validation (3 mRNAs, 3 miRNAs), pathway and functional analysis, the authors look at integrated analysis using miRWalk2.0. Overall, the data derived from this work is of interest to the field and would be suitable for publication in JPM. However, there a few major concerns and many minor concerns this reviewer has before this manuscript could be in an acceptable format.

Major concerns:

  1. This ties to many minor concerns. Although overall, the MS is an okay read, it is plagued by typos, syntax errors, and spelling mistakes. There are too many to mention in the minor concerns so the authors need to carefully edit the MS before resubmission.

Thank you for your suggestion and we have corrected typos, syntax errors and spellings throughout the manuscript. We have also tried to present better resercah design, methods, and  results with conclusions.

  1. Scientifically, the big issue I have is the crux of the novelty of this paper is tying or correlating changes in miRNA to the changes in mRNA observed in autism LCLs. Yet this is a footnote in the results tied to a supplemental table. The authors need to provide a main figure that describes the integrated analysis findings, if even for the most consistent correlations between miRNA changes and mRNA changes.

Thank you for your suggestion, we have added in manuscript Figure 8 and the examples of dysregulated genes-GABRA4, IL27, PTEN, FOXP1, NTN1 and NCAM2.  Also in discussion explained it better.

--- ASD LCLs show upregulation of GABRA4, IL27 and PTEN; whereas downregulated genes were FOXP1, NTN1 and NCAM2 (Figure 8 and Table S7). For example, GABRA4 was 606-fold upregulated in ASD LCLs and regulate pathways of epilepsy and neurodevelopmental disorders (Figure 8, Table S7). GABRA4 was predicted to be targeted by 16 upregulated miRNAs and 10 downregulated miRNAs, respectively. (Figure 8, Table S7). Whereas, only one upregulated miRNA (miR-3529-3p) has been validated to be the target of GABRA4 (Table S7). Table S7 also shows miR-3529-3p was 2 fold upregulated (red color) in ASD LCLs, and it has been validated by 2 publications. PTEN regulate pathways of cell death of immune cells and proliferation of neuronal cells. PTEN was targeted by miR-21-5p, which was 4-fold upregulated in ASD LCLs, and has been validated by 62 publications (Table S7). miR-3529-3p also target NCAM2, which was 6.06-fold upregulated (Figure 8). Sequencing results show miR-3529-3p was 2-fold upregulated in ASD LCLs and only one publication has validated this result (Table S7).

ASD LCLs show the pathways of cell death of immune cells are also activated by IL27, PTEN, and FOXP1 (Figure 8), whereas PTEN, NTN1 and NCAM2 regulate the pathway of proliferation of neuronal cells.

---The above information is shown on lines 306-337;

The figure 8 is as shown

We have expanded and explained this interaction in Discussion also, see lines and 410-445.

Using integrated analysis, follwing RNA-Seq and functional enrichment, we have detected a series of potential miRNA-mRNA interactions with implications in regulation of diverse networks. For example, PTEN was down-regulated by RNA-Seq analysis (Table S3) and by RT-qPCR (Figure 4C). PTEN silencing enhances neuronal proliferation and differentiation by activating PI3K/Akt/GSK3β pathway [39]. . PTEN regulates pathway for cell death of immune cells and proliferation of neuronal cells (shown in red). Table S7 also shows that PTEN is predicted to be targeted by 15 up-regulated miRNAs such as miR-221-5p, miR-21-5p, miR-148b-3p, miR-26b-5p and there are 5 validated targets such as miR-21-5p, miR-152-3p, miR-26b-5p, miR-103a-3p and miR-107. For example miR-21-5p was upregulated in ASD LCLs by 4-fold (Table S7). GABRA4 showed 606.95 Fold upregulation in ASD LCLs (Figure 2, Table S7), and is a target for miR-3529-3p. A role for GABRA4 has been shown for ASD and seizure susceptibility [40, 41]. Integrated analysis shows it regulates the pathway for epilepsy or neurodevelopmetal diosrders , it is predicted to be targeted by 10 miRs such as miR-150-5p, miR-874-5p, miR-33b-3p, miR-940, miR-17-3p, miR-324-5p, miR-766-3p, miR-484, miR-197-3p and miR-342-3p. Similarly for the up-regulated gene IL27 (Figure 2, Table S3) has important neuroprotective role [23] and integrated analysis show it is important in cell death of immune cells has one predicted target miR-7-5p and has no validated miRNA.  Multiple studies have converged on dysregulation of the Wnt/β-Catenin pathway in association with ASD [Caracci et al 2021] including mouse [Platt et al 2017] and Drosophila and zebrafish [Marcogliese et al 2022] animal models and patient cases [Liu et al 2020].

The integrated analysis of miRNA-mRNA showed that 29 miRNAs (18-up-regulated and 11 down-regulated) and 267 genes formed miRNA-target gene pairs, which may be involved in complex network to regulate proliferation of neuronal cells, cell death of immune cells, epilepsy or neurodevelopmental disorders, and Wnt/β-catenin , and PTEN signaling. Table S7 represent new mediators of abnormal gene expression and could be potential targets for further explorations and therapeutic interventions in proliferation of neuronal cells, epilepsy or neurodevelopmental disorders, and Wnt/β-catenin,  and PTEN signaling. However, functional validations are needed to test our miRNA-mRNA integration findings

  1. Related to point 2. I am not buying randomly validating 3 hits. These could be supplemental if adding the above figure is an issue. The hits that should be validated should be based on the integrated analysis to validate the 3 most convincing miRNA – mRNA gene pairs.

Thank you for your comment, and it is very valid issue you brought up. The RNA-Seq experiments were done in 2017 by LC Sciences (see attached excel sheets Exhibit 1 and 2). When we initially decided to validate the miRNA and mRNA expression in later part of 2017, we did not think of doing integrated analysis. This came much later in the write up in 2022. We could have provided additional data, but we ran out of miRNA samples from these experiments.

 We have done recently experiments twice with old cDNA from LCLs RNA experiment for GABRA4 and IL27. WE did not detect IL-27 using TaqMan assay. But for GABRA4 out of 4 ASD and 4 Control samples, we did not get good results as the samples are old. But for your information, the results are shown below: CD2, 3, 5 and 11 are Control LCLs; and CD49, 51, 52 and 53 are ASD LCLs. For ASD LCLs two samples  do support the RNA-seq results that GABRA4 is upregulated.

GAPDH

GABRA4

GAPDH

IL27

CD2

19.1756325

Undetermined

Undetermined

Undetermined

Recently, wD2

19.7045555

Undetermined

Undetermined

Undetermined

CD2

19.7260513

Undetermined

Undetermined

Undetermined

CD3

19.6018944

Undetermined

Undetermined

Undetermined

CD3

20.2670689

Undetermined

Undetermined

Undetermined

CD3

20.9153843

Undetermined

Undetermined

Undetermined

CD5

19.3743973

Undetermined

Undetermined

Undetermined

CD5

19.8086472

37.52714157

Undetermined

Undetermined

CD5

19.9815826

37.35534668

Undetermined

Undetermined

CD11

19.5522766

Undetermined

Undetermined

Undetermined

CD11

18.9151173

Undetermined

Undetermined

Undetermined

CD11

19.3559113

Undetermined

Undetermined

36.67190933

CD49

18.7275162

Undetermined

Undetermined

Undetermined

CD49

18.840704

Undetermined

Undetermined

Undetermined

CD49

17.9092407

Undetermined

Undetermined

Undetermined

CD51

19.7754631

23.17963409

Undetermined

37.75564575

CD51

19.7091408

23.12532806

Undetermined

33.24412918

CD51

20.3989296

23.50553322

Undetermined

32.46919632

CD52

19.403223

Undetermined

Undetermined

34.8568573

CD52

20.1736698

Undetermined

Undetermined

34.84986115

CD52

20.1981678

Undetermined

Undetermined

35.58552551

CD53

19.4834785

30.32563972

Undetermined

Undetermined

CD53

19.6380043

36.28133774

30.33076477

34.52809906

CD53

19.223732

36.25948334

30.82191849

35.0022316

Also for miRNA experiments we are out of cDNA, so we could not run these experiments.

  1. There is a wide inconsistency throughout the paper when referring to AKT/PTEN/WNT pathways:

“PI3K/AKT, ATM and PTEN”

“ATM signaling, PI3K/AKT signaling”

“PI3K/Akt/GSK3β pathway”

“WNT/PTEN signaling”

Please be consistent throughout and see spelling issues in minor concerns. T

 Sorry for any confusion.  These pathways are sometime described together or separate in referenced literature and in pathway analysis, so the description matches how the referenced citation or the pathway analysis has described it.

We agree that inconsistency across the scientific literature is confusing. However, I would agree that WNT and PTEN should be separated in the text and not combined. This has been corrected throughout the manuscript.

  1. The authors spend a large focus on section 4.4 and cancer genes. The section is fine but please heavily condense and make more concise.

Thank you for pointing this out, and we have shortened it per your suggestion as:

Autism and Cancer Genes link? See  Lines 446-495

Some have observed an overlap between the molecular pathways in ASD and cancer [42]. In this study we found many cancer genes that demonstrated changes in both mRNA and miRNA expression in the ASD LCLs. The ASD LCLs demonstrated downregulation of mRNA of several tumor suppressor genes, including KREMEN1[46], ST5 [51], ILDR1[53], and FOXP4 [43].

The ASD LCLs demonstrated upregulation of several oncogenes including SYK [54][56][57], MFHAS1 [58][59][60][61] and TCL1A [64]. Interestingly, MFHAS1 has also been implicated in sepsis-associated encephalopathy [62] and intellectual impairment [63]. In contrast, SSBP2, a gene in involved in DNA and telomere repair and growth arrest of cancer cells [65], was also found to be upregulated in ASD LCLs.

Several miRNAs involved in cancer show deregulated expression in ASD LCLs, including miR-6732-3p, hsa-miR-221-5p [66] [67], hsa-miR-21-5p [68] [69], hsa-miR-152-3p[70] [71], hsa-miR-3529-3p [72][73],and  hsa-miR-148b-3p [74][75]. Interestingly, miR-3529-3p which targets GABRA4 shows its expression levels are decreased in the serum of liver cancer patients [Weng et al 2021].

  1. Wnt is an important pathway in neurodevelopmental disorders/autism. The authors should cite some literature showing increased and decreased Wnt is ASD related genes/conditions (see. PMID: 31981384, PMID: 35044823, PMID: 28402856).

Thank you for your suggestion. We have included this in the discussion. At lines 433-436.and reads as:

Multiple studies have converged on dysregulation of the Wnt/β-Catenin pathway in association with ASD [Caracci et al 2021] including mouse [Platt yet al 2017] and Drosophila and zebrafish [Marcogliese et al 2022] animal models and patient cases [Liu et al 2020].

  1. The beginning of the discussion, rightfully, compares the authors results to other expression analysis of ASD cells. The authors need to comment in section 4.1 on the overall discrepancies and potential limitations of performing these types of experiments.

The  limitations of the investigation has been added at line 377-385 and reads as:

Many studies, including this one, have many limitations, particularly the modest samples sizes as compared to the heterogenous nature of ASD. Indeed, ASD defined by a collection of various symptoms which can be different from patient to patient. Although some categories are used as modifiers of the diagnosis such as with and without language impairment or intellectual disability, the variation in symptoms has been difficult to easily describe or categorize. Thus, without well-defined subgroups it may be difficult to find consistency with modest sample sizes. Clearly larger studies on well-defined groups of individuals with ASD will be needed in the future. However, despite the heterogenous nature of ASD this study did find consistency with some previous studies.

  1. In section 4.3 lines 328 to 331, the authors are referring to results, even stating shown in read without referring to a figure. Please re-write this section more clearly.

Thank you for your suggestion. We have corrected this information and cited the Figure 4C for PTEN;  Now it reads?

For example, PTEN was down-regulated by RNA-Seq analysis (Table S3) and by RT-qPCR (Figure 4C). PTEN silencing enhances neuronal proliferation and differentiation by activating PI3K/Akt/GSK3β pathway [39]. See lines 412-414.

Minor concerns:

- Every gene mentioned needs to be italicized, unless you are talking about protein level, which I don’t think the authors ever are.

Thank you for your suggestion, we have italicized the gene symbols.

-Page 2 lined 45 – “beta-catherin”, I assume should read beta-catenin, I also recommend using β if allowed by the journal as well as being consistent in later instances

Thank you for pointing out this error, we have corrected it.

- this reviewer does not understand the repeated instances of “Epilepsy / epilepsy”. Please either correct, explain, or simplify with “seizure-disorders”. This shows up a few times.

This was referring to epilepsy and the repeats have been corrected.

-Page 3 line 76, remove space between CA and closed bracket.

Thank you for pointing out this error, we have corrected it.

-Page 3 line 77, first word should be plural, “Samples”

Thank you for pointing out this error, we have corrected it.

- Page 3 line 82, spacing issues after “were”

Thank you for pointing out this error, we have corrected it.

-Page 4 line 131, “. (Fig. 1).” Remove first period, and other figures are referred as “Figure X”. Please make consistent.

Thank you for pointing out this error, we have corrected it.

- Page 6 line 180, remove space between Figure 3 and open bracket.

Thank you for pointing out this error, we have corrected it.

- Page 7 line 202, please unitalicized “and”.

Thank you for pointing out this error, we have corrected it.

- figure 6, I am unsure what Wnt/I2-catenin signaling is. Should read Wnt/β-catenin. I also recommend this to be used for most instances in the paper.

Thank you for this suggestion. We have changed this in the figure 6 and 7 as Wnt/β-catenin signaling .

- Page 10 line 256, fix double periods

Thank you for pointing out this error, we have corrected it.

- Page 10 line 258, fix “signaliing” to signaling

Thank you for pointing out this error, we have corrected it.

- Page 11 line 286, fix spacing issue after “results”

Thank you for pointing out this error, we have corrected it.

- Page 11 line 298, fix “upregulaation” to upregulation

Thank you for pointing out this error, we have corrected it.

- Page 11 line 299, fix “cntrol” to control

Thank you for pointing out this error, we have corrected it.

- Page 11 line 307 and 310, fix “abarrent” to aberrant

Thank you for pointing out this error, we have corrected it.

- Page 12 line 313, assuming “th” should read “that”?

Thank you for pointing this out. It reads now: IL-27 is a pleiotropic cytokine involved in infection, cellular stress, neurological disease and cancer, has complex activating and inhibitory properties in both innate and acquired immunity [34].

- Page 12 line 318, please explain or expand BTBR

BTBR is the name of one of the standard genetic models of ASD. The full name is the BTBR T+Itpr3tf/J mouse because of the mouse has a deletion in the Itpr3 gene but the standard name for this mouse in the literature is the BTBR mouse.

- Page 12 line 318, assuming “mode” should read “model”?

Thank you for pointing out this error, we have corrected it.

Again, this is not all the issues, there are more, please spell and grammar check closely before resubmitting.

Thank you for pointing out the errors and we will run spell check before resubmitting,

we have corrected it.

Reviewer 3 Report

Gill et al. report the identification of DEGs (mRNAs and miRNAs) in lymphoblastoid cell lines (LCLs) from children with autistic disorder. The research follows a logical flow, going from the identification of differentially expressed mRNAs and miRNAs to the enrichment of those DEGs in pathways and their possible interactions, founding some interesting genes and pathways. A validation with qRT-PCR it’s also included. However, some issues should be addressed before the manuscript is suitable for publication. My detailed comments are as follows:

Mayor comments

  • Material and methods: I find having part of the methods in the supplementary not convenient, I would include it in the main text.
    • In section 2.4 it's indicated that reads were aligned using bowtie and then Read counts were estimated with RSEM whereas in Supplementary Methods HISAT and BLAST are mentioned as aligners and a different workflow. This should be clarified and unified. I also find it important that self-scripts are made available to improve the reproducibility of the findings (through Github for example).
    • As an FDR correction has been performed, I don't find it useful to sometimes use all DE genes with pValue<0.05 and sometimes use FDR<0.05. I think all analyses should include only genes with FDR<0.05. Also important will be to unify pValue and FDR along the paper to make more clear when it's FDR corrected or not.
  • Data availability: A major flaw is the dataset has not been made available. the RNA-seq data should be deposited in NCBI's SRA database or NCBI's Gene Expression Omnibus (GEO).

Minor comments:

  • Line 49 and 57: miRNA -> miRNAs
  • Figure 1: significant genes/miRNAs -> differentially expressed genes/miRNAs (or something similar that indicates that is a DE analysis)
  • Citation of some bioinformatics tools and DBs is missing:
    • Line 100: bowtie
    • Line 101: RSEM
    • Line 102: DEseq2 and Ensembl
    • Supplementary: FastQC, gffcompare, miRbase, BLAST, RNAfold
  • In Methods S2: different -> differential
  • Line 128: was employed for employed -> was employed
  • Line 155: The samples with poor alignment rates were dropped from the further analysis -> This to the methods section.
  • Line 158: A false positive rate of alfa=0.05 with false discovery rate (FDR) correction... -> A false discovery rate (FDR) of alfa=0.05...
  • Line 186: The down-regulated -> The top down-regulated
  • Line 196: sequencing profiling results -> sequencing results
  • SUGGESTION: In Section 3.5 a figure will help to show the relevancy of these results
  • Line 263: differntially -> differentially
  • Line 399: RNA-seq, analysis -> RNA-seq analysis

Author Response

Mayor comments

  1. Material and methods: I find having part of the methods in the supplementary not convenient, I would include it in the main text.

Thank you for your suggestion on methods and we will add into the main manuscript body. See lines 108-156.

  1. In section 2.4 it's indicated that reads were aligned using bowtie and then Read counts were estimated with RSEM whereas in Supplementary Methods HISAT and BLAST are mentioned as aligners and a different workflow. This should be clarified and unified. I also find it important that self-scripts are made available to improve the reproducibility of the findings (through Github for example)

Thank you for this suggestion.

We have clarified the bioinformatics analysis and added necessary references on

for the above listed software’s and analysis in the methods section: lines 139-149

It reads as:

The raw reads of both sequencing profiles were aligned using bowtie [Langmead et al 2009] against hg19 version of the human genome, and RSEM v1.2.12 software [Li and Dewey 2011] was used to estimate raw read counts using Ensemble v84 gene information. DESeq2 [Love et al 2014] was utilized to determine the significance of differential expression between sample groups. A false-positive rate of α = 0.05 with false discovery rate (FDR) correction was taken as the level of significance. Only a handful of genes were found to satisfy the FDR<5% cut-off which were not sufficient for a functional enrichment analysis. Therefore, it was decided to consider a p<0.05 threshold to select differentially expressed genes. These genes were then subjected to functional enrichment analysis.

.

RNA-seq work was done in 2017 by LC Sciences. We tried to deposit the data at GEO, but as per GEO the fastq files are defective (see exhibit 2). We will provide GEO link once it is deposited, as we are waiting from LC Sciences to provide correctly formatted fastq files.

Once we have the correct formatted fastq files-we will have raw sequencing deposited and available under controlled access NCBI's Gene Expression Omnibus (GEO). We will provide the RNA-Seq data accession code: XXXXXX

  1. As an FDR correction has been performed, I don't find it useful to sometimes use all DE genes with pValue<0.05 and sometimes use FDR<0.05. I think all analyses should include only genes with FDR<0.05. Also important will be to unify pValue and FDR along the paper to make more clear when it's FDR corrected or not.

Thank you for this suggestion.

Initially, a false-positive rate of α = 0.05 with false discovery rate (FDR) correction was taken as the level of significance to select differential expressed genes and miRNA. However, in case of genes, there were only handful of candidates satisfied the FDR cutoff which were insufficient to carry out functional enrichment analysis. Therefore, we selected all the genes with p<0.05 as significant candidates and then these genes were considered for pathways enrichment analysis.

See lines 140-149.

  1. Data availability: A major flaw is the dataset has not been made available. the RNA-seq data should be deposited in NCBI's SRA database or NCBI's Gene Expression Omnibus (GEO).
    • .

Thank you for your suggestion. This data was analyzed in the middle of 2017 by LC Sciences (see attached Exhibit 1 and 2). We are in process of depositing the data in GEO. Initial attempt showed some problems with the files (see attached Exhibit 3), and we are awaiting correctly formatted fastq files from LC Sciences.

Once we have the correct formatted fastq files-we will have raw sequencing deposited and available under controlled access NCBI's Gene Expression Omnibus (GEO). We will provide the RNA-Seq data accession code: XXXXXX

Minor comments:

  • Line 49 and 57: miRNA -> miRNAs
  • Thank you for pointing out this error, we have corrected it.

  •  
  • Figure 1: significant genes/miRNAs -> differentially expressed genes/miRNAs (or something similar that indicates that is a DE analysis)

Thank you for your suggestion, we have changed it to differentially expressed genes/miRNAS. See lines 715-78.

Citation of some bioinformatics tools and DBs is missing:

  • Line 100: bowtie
  • Line 101: RSEM
  • Line 102: DEseq2 and Ensembl
  • We have added the required references

  • Li B, Dewey CN. RSEM: accurate transcript quantification from RNA-Seq data with or without a reference genome.BMC Bioinformatics. 2011;12:323. doi: 10.1186/1471-2105-12-323.
  •  
  • Love, M. I., Huber, W., Anders, S. Moderated estimation of fold change and dispersion for RNA-seq data with DESeq2.Genome Biology. 15, 550 (2014).

  • Supplementary: FastQC, gffcompare, miRbase, BLAST, RNAfold
  • In Methods S2: different -> differential
  • Thank you for this suggestion, we have corrected it in the manuscript methods section.
  •  
  • Line 128: was employed for employed-> was employed T
  • Thank you for this suggestion, we have corrected it.
  • Line 155: The samples with poor alignment rates were dropped from the further analysis-> This to the methods section. Harsh and Priyankara
  •  
  • Line 158: A false positive rate of alfa=0.05 with false discovery rate (FDR) correction...-> A false discovery rate (FDR) of alfa=0.05... . Harsh and Priyankara
  •  
  • Line 186: The down-regulated-> The top down-regulated . Harsh and Priyankara
  • .
  • Line 196: sequencing profiling results -> sequencing results
  • Thank you for this suggestion, we have corrected it.
  1. SUGGESTION: In Section 3.5 a figure will help to show the relevancy of these results
  2.  
  3. Thank you for your suggestion, we have added in manuscript Figure 8 and the examples of dysregulated genes-GABRA4, IL27, PTEN, FOXP1, NTN1 and NCAM2. Also in discussion explained it better.
  4.  
  5. --- ASD LCLs show upregulation of GABRA4, IL27 and PTEN; whereas downregulated genes were FOXP1, NTN1 and NCAM2 (Figure 8 and Table S7). For example, GABRA4 was 606-fold upregulated in ASD LCLs and regulate pathways of epilepsy and neurodevelopmental disorders (Figure 8, Table S7). GABRA4 was predicted to be targeted by 16 upregulated miRNAs and 10 downregulated miRNAs, respectively. (Figure 8, Table S7). Whereas, only one upregulated miRNA (miR-3529-3p) has been validated to be the target of GABRA4 (Table S7). Table S7 also shows miR-3529-3p was 2 fold upregulated (red color) in ASD LCLs, and it has been validated by 2 publications. PTEN regulate pathways of cell death of immune cells and proliferation of neuronal cells. PTEN was targeted by miR-21-5p, which was 4-fold upregulated in ASD LCLs, and has been validated by 62 publications (Table S7). miR-3529-3p also target NCAM2, which was 6.06-fold upregulated (Figure 8). Sequencing results show miR-3529-3p was 2-fold upregulated in ASD LCLs and only one publication has validated this result (Table S7).
  6. ASD LCLs show the pathways of cell death of immune cells are also activated by IL27, PTEN, and FOXP1 (Figure 8), whereas PTEN, NTN1 and NCAM2 regulate the pathway of proliferation of neuronal cells.
  7.  
  8. ---The above information is shown on lines 306-337;
  9. The figure 8 is as shown
  10.  

We have expanded and explained this interaction in Discussion also, see lines and 410-445.

Using integrated analysis, follwing RNA-Seq and functional enrichment, we have detected a series of potential miRNA-mRNA interactions with implications in regulation of diverse networks. For example, PTEN was down-regulated by RNA-Seq analysis (Table S3) and by RT-qPCR (Figure 4C). PTEN silencing enhances neuronal proliferation and differentiation by activating PI3K/Akt/GSK3β pathway [39]. . PTEN regulates pathway for cell death of immune cells and proliferation of neuronal cells (shown in red). Table S7 also shows that PTEN is predicted to be targeted by 15 up-regulated miRNAs such as miR-221-5p, miR-21-5p, miR-148b-3p, miR-26b-5p and there are 5 validated targets such as miR-21-5p, miR-152-3p, miR-26b-5p, miR-103a-3p and miR-107. For example miR-21-5p was upregulated in ASD LCLs by 4-fold (Table S7). GABRA4 showed 606.95 Fold upregulation in ASD LCLs (Figure 2, Table S7), and is a target for miR-3529-3p. A role for GABRA4 has been shown for ASD and seizure susceptibility [40, 41]. Integrated analysis shows it regulates the pathway for epilepsy or neurodevelopmetal diosrders , it is predicted to be targeted by 10 miRs such as miR-150-5p, miR-874-5p, miR-33b-3p, miR-940, miR-17-3p, miR-324-5p, miR-766-3p, miR-484, miR-197-3p and miR-342-3p. Similarly for the up-regulated gene IL27 (Figure 2, Table S3) has important neuroprotective role [23] and integrated analysis show it is important in cell death of immune cells has one predicted target miR-7-5p and has no validated miRNA.  Multiple studies have converged on dysregulation of the Wnt/β-Catenin pathway in association with ASD [Caracci et al 2021] including mouse [Platt et al 2017] and Drosophila and zebrafish [Marcogliese et al 2022] animal models and patient cases [Liu et al 2020].

The integrated analysis of miRNA-mRNA showed that 29 miRNAs (18-up-regulated and 11 down-regulated) and 267 genes formed miRNA-target gene pairs, which may be involved in complex network to regulate proliferation of neuronal cells, cell death of immune cells, epilepsy or neurodevelopmental disorders, and Wnt/β-catenin , and PTEN signaling. Table S7 represent new mediators of abnormal gene expression and could be potential targets for further explorations and therapeutic interventions in proliferation of neuronal cells, epilepsy or neurodevelopmental disorders, and Wnt/β-catenin,  and PTEN signaling. However, functional validations are needed to test our miRNA-mRNA integration findings

  • Line 263: differntially-> differentially
  • Thank you for this suggestion, we have corrected it.
  •  
  • Line 399: RNA-seq, analysis-> RNA-seq analysis
  • Thank you for this suggestion, we have corrected it.
  •  

Round 2

Reviewer 2 Report

This reviewer understands the limits of working with cDNA from human cells. The authors have addressed the majority of my concerns. The manuscript is suitable for publication.